# Uniformity Attentive Learning-Based Siamese Network for Person Re-Identification

**DOI:** 10.3390/s20123603

**Published:** 2020-06-26

**Authors:** Dasol Jeong, Hasil Park, Joongchol Shin, Donggoo Kang, Joonki Paik

**Affiliations:** Department of Image, Graduate School of Advanced Imaging Science, Multimedia and Film, Chung-Ang University, Seoul 06974, Korea; jds2953@cau.ac.kr (D.J.); hahaha2470@cau.ac.kr (H.P.); mbstel@cau.ac.kr (J.S.); tiruss@cau.ac.kr (D.K.)

**Keywords:** person re-identification, attention mechanism, Siamese network

## Abstract

Person re-identification (Re-ID) has a problem that makes learning difficult such as misalignment and occlusion. To solve these problems, it is important to focus on robust features in intra-class variation. Existing attention-based Re-ID methods focus only on common features without considering distinctive features. In this paper, we present a novel attentive learning-based Siamese network for person Re-ID. Unlike existing methods, we designed an attention module and attention loss using the properties of the Siamese network to concentrate attention on common and distinctive features. The attention module consists of channel attention to select important channels and encoder-decoder attention to observe the whole body shape. We modified the triplet loss into an attention loss, called uniformity loss. The uniformity loss generates a unique attention map, which focuses on both common and discriminative features. Extensive experiments show that the proposed network compares favorably to the state-of-the-art methods on three large-scale benchmarks including Market-1501, CUHK03 and DukeMTMC-ReID datasets.

## 1. Introduction

Person re-identification (Re-ID) is an important research topic in computer vision. The objective of person Re-ID is to retrieve a specific person from multiple, non-overlapping camera views in a multi-camera surveillance system. Given a target person (query) in a specific camera view, it is matched with persons of interest (gallery) in different camera views. For that reason, person Re-ID has widely been applied to video analytics applications such as multi-target tracking [1] and human retrieval [2].

Traditional person Re-ID tasks commonly used distance metric learning [3,4] and hand-crafted features [5,6,7], which are mainly based on color and salient edge histogram [7,8,9]. The traditional approach used image segmentation to generate salient edges of body shape and computed the HSV color histogram. Zhao et al. applied SIFT descriptor to densely extract distinctive feature patches and create the La*b* color histogram. Many hand-crafted feature-based works [3,4,10,11] have employed distance metric learning such as unsupervised clustering, nearest neighbors, or kernel-based classifiers for computing the sparse pairwise similarity. These methods tend to make similar data points closer to each other than dissimilar points.

As deep learning evolves, Re-ID has significantly improved on the widely used benchmarks [12,13,14]. Early deep Re-ID methods used global feature representation learning to apply image classification [15,16,17,18,19,20] into the Re-ID task [13]. However, the naive classification networks cannot solve the fundamental problem of Re-ID including: background clutter, occlusion and large intra-class variation.

Attention schemes focus on the deterministic region to accurately match a query with gallery. As shown in Figure 1a, the first and second rows are the same person, but the bag is missing in the second row. In this case, the bag is not a distinctive feature. On the other hand, as shown in Figure 1b, the bag disappeared in another camera view, where the bag is not a distinctive feature either. However, the proposed activation maps of (a) and (b) focused on the shoes and shirts, respectively. In the first row, the bag is considered important feature, but not in another camera views. Therefore, it is necessary to focus on common and discriminative features in consideration of the characteristics of Re-ID. We have achieved this goal through uniformity loss that minimizes differences in the attention map of objects. The main contributions of this work are summarized as follows:We proposed an attentive learning-based Siamese network for person Re-ID. Our method includes a channel attention and encoder-decoder attention modules for robust feature extraction.We proposed uniformity loss for learning both common and discriminative features. The proposed loss helps the Siamese network to learn more important features accurately.Extensive experiments conducted on three common benchmarks show that the proposed method achieves comparable results in terms of both subjective and objective measures.

This paper is organised as follows. In Section 2, we describe the related works of the proposed nework. The proposed uniformity attentive learning-based Siamese network is presented in Section 3 followed by experimental result in Section 4, and Section 5 concludes this paper.

## 2. Related Works

### 2.1. Attention Mechanism

An attention mechanism generates attentive regions and performs spatial localization using a neural network. Concept of the attention mechanism originated from the study in natural language processing [21,22,23]. Mnih et al. proposed a recurrent neural network (RNN) model [24] that adaptively selects a region of interest from an image or video. However, in the recurrent attention model, it is difficult to focus on a definite point in the image in the training process, which is referred to as hard attention problem. To solve this problem, Bahdanau et al. proposed a soft attention model [25], which automatically searches all input features. It can overcome the limitations of containing all the sentence information in a fixed-length vector in the RNN encoder-decoder network. Bahdanau’s method achieved a significantly improved translation performance in natural language processing.

Recent developments in attention mechanisms have been influenced the various computer vision applications. Xu et al. presented an attention-based image captioning model that automatically learns to describe the content of a given image [26]. Sermanet et al. present an attention model for image classification, which learns to detect high-resolution attention to extract discriminative regions in the image [27]. Li et al. applied global and local contexts to the object detection field through the attention model [28]. Li’s method generates an attention map to highlight global contextual locations and exploits inside and outside local contextual information.

The attention mechanism allowed the Re-ID tasks to overcome the localization and misalignment problems. Liu et al. proposed HydraPlus-Net (HP-net) to apply the attention mechanism to pedestrian analysis [29]. In HP-net, multiple attention networks capture low- to semantic-level features, and selectively explores the multi-scale features to enrich the image representation. Li et al. designed Harmonious Attention CNN (HA-CNN), which jointly learns soft and hard pixel attention to optimize the misaligned images [30]. Li et al. proposed a spatio-temporal attention model to discover a discriminative body parts [31]. This model learns multiple spatial attention models and employs a diversity regularization. However, these methods usually formulate a local part-based approach to solve the spatial localization problem.

### 2.2. Siamese Network for Person ReID

Recently, the Siamese network has gained attraction to predict identification and similarity score for person re-identification [32,33]. Zheng et al. optimized both identification and verification losses to solve inter- and intra-class variation problems, respectively [34]. Cheng et al. proposed an embedding network that consists of multiple channels to learn global and local body-parts features [35]. Cheng’s network used the triplet loss function to make the distance from the positive sample closer and to make the negative samples farther away from each other. Chen et al. presented quadruplet ranking loss, which extended the triplet loss to increase negative samples and another probe with a margin-based hard negative mining strategy [36]. The sample is adaptively selected to the margin threshold according to the trained model. Li et al. constructed a five-channel Siamese model that extracts both global and local features using two branches, and then fuses the verification and identification information from multiple channels [37]. However, these models need an extra local part-based method since they learn the network to extract the features and to estimate distance metrics. For that reason, it cannot immediately solve the spatial localization problem. Guo et al. proposed attention consistency loss for classification problems using two-branch network [38]. This model finds the consistency of attention regions of the same class by learning the distance between the input and the transformed heatmaps. Although attention consistency is useful to extract common features, it is not suitable for finding discriminative features.

Re-ID tasks are important to solve the inter class variation problem, such as distinguishing others in similar clothes. Therefore, we designed attention loss to learn uniformity features using the Siamese property. The uniformity loss not only learns common features, but also distinguishes features from others. In addition, the Siamese property can directly improve the spatial localization for an end-to-end learning. Our architecture inherits the advantages of the latest attention models to extract features that are suitable for Re-ID.

## 3. The Proposed Method

In this section, we present a novel attention learning network for person Re-ID. The proposed method uses the Siamese network to aggregate the identification, verification and uniformity losses in Section 3.1, Section 3.2, and Section 3.3, respectively. As shown in Figure 2, each branch generates the final feature *f* as a feature extractor that is applied to the attention module. *f* returns the predicted value for each loss through additional layers such as global average pooling, fully-connected layer, or spatial attention.

### 3.1. Identification Using Attention Module

The attention module involves an encoder-decoder attention module and channel attention module to generate the discriminative features and spatial attention to concentrate on important features. The identification module extracts the attention region in the form of a feature map of the convolution layer.

#### 3.1.1. Encoder-Decoder Attention Module

Encoder-decoder architecture is used to capture multiple scale information and output pixel-wise predictions [39]. Newell et al. combined multiple-scale information by repeatedly stacking this structure, called the hourglass. The encoder lowers the resolution of the input to obtain semantic information, and the decoder up-sampling to the original input size. We can combine output and input of this structure to get multiple-scale information simultaneously. Our network applied this structure at the high-level layer to understand the whole body. The encoder processes convolutional layers for the feature to reach low-resolution. The decoder can learn weights by performing deconvolutional layer for up-sampling. The encoder-decoder attention branch consists of three convolution layers and three deconvolution layers. This attention module helps to remove the background clutter and see about the shape of whole body by creating a soft attention mask. As shown in Figure 3, the encoder path repeats unpadded convolution and ReLU three times. The decoder path repeats deconvolution and ReLU three times, and the final deconvolution reduces the channel to 1.

#### 3.1.2. Channel Attention Module

In the channel attention branch, we flatten the feature map to understand the relationship between channels without spatial information. In other words, we generate a feature vector x using global average pooling to flatten F. In Equation (Equation 1), x is generated by dividing the c-th channel of F by the size of each channel. The channel attention operator, denoted as $f_{ch}$, consists of a dimension reduction layer, activation layer (ReLU), dimension expansion layer and sigmoid function. We use the sigmoid function to give the importance to each channel to generate the channel attention feature vector $A_{ch}\in R^{c\times 1\times 1}$.
(1)$$x_{c}=\frac{1}{H\times W}\sum_{i=1}^{H}\sum_{j=1}^{W}F_{c}\left(i,j\right)$$
(2)$$A_{ch}=f_{ch}\left(x,W\right)=\sigma \left(W_{2}\delta \left(W_{1}x\right)\right)$$
where σ denotes the sigmoid function, δ is the ReLU function. $W_{1}\in R^{\frac{C}{r}\times C}$ and $W_{2}\in R^{C\times \frac{C}{r}}$ respectively representation the dimension reduction and expansion layers with reduction ratio *r*. We used r=16 as the experimentally best reduction ratio. We then multiply the feature map to the encoder-decoder attention map and channel attention vector to obtain the final feature map $\hat{F}$ defined as:(3)$$\hat{F}=\left(F⊙A_{ch}\right)⊙A_{ed}$$
where ⊙ represents the tensor multiplication operation. As shown in Figure 4, $\hat{F}$ embedded the layer4 of pre-trained ResNet50, and the feature vector *f* predicts identities through the identification classifier.

#### 3.1.3. Identification Loss

The identification is adopted from the work in [40] and a more recent paper that uses ResNet50 in Siamese network [34]. Identification classifier follows layer4, which the proposed feature extractor. The identification classifier with two fully-connected layers generates an identity prediction for the input image. In other words, *f* predicts $\hat{y}$ through the identification classifier. We train the cross-entropy loss for predicted identity label by the softmax function to optimize the identification loss $L_{id}$. The $\hat{y}$ and $L_{id}$ are defined as
(4)$${\hat{y}}_{i}=\frac{w_{I}\circ f_{i}}{\sum_{j}w_{I}\circ f_{j}},$$
(5)$$L_{id}=-\sum_{n=1}^{N}y_{i}\log{\hat{y}}_{i},$$
where $w_{I}$ denotes the parameters of fully-connected layers. $y_{i}$ is *i*-th identity label, and ${\hat{y}}_{i}$ is its prediction.

### 3.2. Siamese Verification Loss

The verification loss [34] is calculated in the Siamese network structure to directly compare the high-level features. In our network, the high-level features represent the output *f* of layer4 as shown in Figure 4. We add a global average pooling and fully-connected layers to output feature vector *z* of size 512×1×1 to compare the two high-level features. From each branch of the Siamese network, $z_{1}$ and $z_{2}$ perform tensor product to obtain similarities. The tensor product is denoted as $z_{m}=z_{1}*z_{2}$. The $z_{m}$ is embedded to a two-dimensional (2D) vector ${\hat{v}}_{i}$ by the softmax function as shown in Equation (Equation 6). Like the binary classification problem, we use binary cross-entropy loss to determine whether two images are the same or not as
(6)$${\hat{v}}_{i}=\frac{z_{m,i}}{\sum_{j=1}z_{m,j}},$$
(7)$$L_{ver}=-\sum_{i=1}^{2}v_{i}\log\left({\hat{v}}_{i}\right),$$
where $v_{i}\in \left\{0,1\right\}$ is verification label, ${\hat{v}}_{i}$ is the correspondingly predicted output. If two input images are expected to be the same person, $v_{i}=1$, otherwise $v_{i}=0$.

### 3.3. Attention Uniformity Loss

Only the identification attention module cannot generate a uniform attention, even in the same person image. In the second column of in Figure 1a, the first row focuses on the bag. However, if the bag is invisible in another camera view, it may make a wrong decision. In the second column of Figure 1b concentrate on the local region. When compared to others, it is difficult to distinguish similar characteristics. To solve these problem, attention maps should be focused on features that are distinct from other people and common to the same person.

A correlation method is popular to compare the similarity between two images using Siamese network. However, it is not suitable for the Re-ID task with a large intra-class variation and small inter-class variation. Therefore, we propose uniformity loss that uses the property of the Siamese network by simultaneously considering features of Re-ID. The uniformity loss uses the attention score calculated in the attention module of each branch. The attention score is adjusted using the spatial attention module from the final feature maps. Spatial attention module compresses the channel dimension to concentrate the spatial information. The $\hat{F}$ in Equation (Equation 3) is fed to layer4 and outputs $f\in R^{C\times H\times W}$. We compute the attention score by reducing the channel *C* to 1 from *f* using a 1×1 convolution layer. The attention score is a tensor, $s\in R^{1\times H\times W}$.

For the robust attention uniformity, we use the triplet loss [41] that minimizes the distance of intra-classes and maximizes the distance of inter-classes. For different images, $\left(I_{q},I_{p},I_{n}\right)$, $\left(I_{q},I_{p}\right)$ is pair of same class, and $\left(I_{q},I_{n}\right)$ is not. We define the attention uniformity loss as:(8)$$L_{unif}\left(s_{q},s_{p},s_{n}\right)=\max\left(d\left(s_{q},s_{n}\right)-d\left(s_{q},s_{p}\right)+m,\;0\right)$$
where $s_{q},s_{p},s_{n}$ respectively represent attention scores of $\left(I_{q},I_{p},I_{n}\right)$ through the embedding network, *d* represents the $l_{1}$-distance, and *m* is the margin between the intra- and inter-class to distinguish similar people. This loss can extract common and discriminatory features.

### 3.4. Overall Architecture and Final Loss

The proposed Siamese network-based model consists of two convolution layers with shared weights and attention module. Each branch of Siamese network learns the weights that are extracted features for identification of each input images. As shown in Figure 2, each convolution layer shares weights. In the Siamese structure, the extracted feature vector is a high-level feature, and is directly calculated as verification prediction and attention uniformity without ReLU. We optimize identification, verification and uniformity losses as the final loss function as:(9)$$L=L_{id}+L_{ver}+L_{unif}$$
where $L_{id}$ refers the identification loss, $L_{ver}$ is the verification loss, and $L_{unif}$ is attention uniformity loss. The proposed network architecture are detailed in Table 1 using ResNet50 as the baseline network [20].

## 4. Experiments

### 4.1. Datasets and Evaluation Metrics

**Dataset.** We evaluated three large-scale person Re-ID benchmarks including Market1501 [12], CUHK03 [13,42] and DukeMTMC-reID [14]. Market1501 has 32,668 person images with 1501 identifications from 6 camera views. Its train set has 12,936 images with 751 identities. Query and gallery sets have 3368 and 19,732 images, respectively with 750 identities for testing. CUHK03 collects 1467 identities to provide manually labeled and auto-detected bounding boxes from 14,096 and 14,097 images, respectively. We adopt a new training and testing split protocol into 767 and 700 identities. DukeMTMC-reID is a subset of the DukeMTMC dataset [43] for image-based re-identification with 1402 identities and 36,411 images. We utilize the standard training/testing split equllery into 702 identities.

**Evaluation Metrics.** We use the cumulative matching characteristic (CMC) and mean average precision (mAP) metrics on all datasets. The CMC curve records the actual match within the top n-ranks, while the mAP evaluates the overall performance of the method considering the precision.

### 4.2. Implementation Details

We employed pre-trained ResNet50 [20] on ImageNet [44] as the basic backbone. We stacked the attention module on the third residual block of ResNet50. We embedded network by removing the existing fully-connected layer, and appended linear blocks of dimension 512 to produce the feature vector *z*. The feature vector *z* is used to predict a label through a fully-connected layer. During training, *z* is also used to compare a pair of images. Each batch is composed of ramdomly selected *P* identities, and a pair of positive and negative samples is evenly selected for each *P*. For verification, the feature vector is extracted by embedding the network for each pair and is compared with the feature vector of each *P*. In addition, the attention score of *P* identities and each pair are calculated as L1-norm to optimize uniformity loss. All person images are resized to 256×128. For training, the stochastic gradient descent (SGD) alorithm is used with initial learning rate 0.04, decayed by 0.1 at every 30 epoch and momentum 0.9. During the test phase, our network only involves the embedding network and processes only for the *P* identities of test batch without images pair. Our network is implemented in the PyTorch framework with two NVIDIA GTX 1080Ti GPUs. It took about 4 hours to train the models on the Market-1051 dataset. This set of parameters was used for all three datasets in our experiments. Our model has 27.7 million parameters, the number of floating-point operations (FLOPs) is $2.90\times 10^{9}$. In the test phase, it will take 0.02 seconds per image, which is approximately equivalent to 50 frames per second (fps).

### 4.3. Comparison with State-of-the-Art Method

**Market1501** We achieved a competitive performance as shown in Table 2. Our method (ResNet50) and second best method HA-CNN [30] have similar performance by at least +8.9% Single Query (SQ)/+7.0% Multi Query (MQ) rank-1 accuracy and +18.2% (SQ)/+2.5% (MQ) mAP improvements than third best method MSCAN [45], respectively.

**CUHK03** We achieved competitive results on both detected and labeled person images as shown in Table 3. The proposed method (ResNet50) substantially outperforms for rank-1 (+6.1%, +1.9%) and mAP (+4.8%, +5.5%) on detected and labeled sets respectively. Comparing with attention based method HA-CNN [30], our method achieved clear gains of +17.2% and +14.0% for rank-1 and mAP on detected set.

**DukeMTMC-reID** We achieved experimental results on DukeMTMC-reID dataset, which is a more challenging Re-ID dataset than Market1501. This benchmark has more intra-class variations in the wider camera view and background clutter. As shown in Table 4, our method (ResNet50) achieved the best results by +0.2% and +2.3% in rank-1 and mAP, respectively.

### 4.4. Ablation Study

#### 4.4.1. Efficiency of the Proposed Attention Module and Uniformity Loss

We further evaluated the effect of the attention module and uniformity loss on Market1501 dataset. As shown in Table 5, “Baseline” represents our baseline schemes with IDE and Verification losses. We trained ResNet50 [20] and VGG16 [16] as the baseline of our network. Each baseline is pre-trained on ImageNet [44]. We added an attention module to each baselines without uniformity loss. It improves the rank-1 accuracy over the ResNet50 and VGG16 by 1.6% and 1.9%, respectively. In ResNet50 with an attention module, the mAP accuracy was improved by 5.5%. The combination of attention module and the corresponding uniformity loss gives higher accuracy than the attention module only scheme. In ResNet50-baseline, it achieved the performance with +1.5% on the rank-1 and +2.2% on the mAP score. Also, in VGG16-baseline, our network obtains 2.0% and 3.1% improvement, respectively.

These result show that our network works on different baselines can improves their performance. Especially, the attention module (AM) significantly improves the mAP score, and the uniformity loss (UL) showed improved performance in both rank accuracy and mAP score.

#### 4.4.2. Comparison on Network Architectural Change

In order to investigate the effect of the proposed uniformity attentive learning-based Siamese network, we conducted an experiment that applied the attention module at different scales of the backbone network. As shown in Table 6, the result of applying the attention module in layer3 was most effective. In order to apply the effective attention module, it is important to select a feature map that properly contains coarse and fine information. Since the receptive fields of layer1 and layer2 are not large enough, the encoder-decoder attention cannot see the whole body shape. Because layer4 has a very small feature map of size 8×4, the attention map of the encoder-decoder attention is not suitable for learning descriminative features. In the encoder of the encoder-decoder attention, the input scale is reduced by 1/4, which is not enough to extract fine information. In this experiment, we removed one convolution layer of the encoder because the feature map size of layer4 is no longer smaller. Layer3 has a sufficiently large receptive field and a feature map of size 16×8 is appropriate. Therefore, common and discriminative features are best learned in layer3.

### 4.5. Qualitative Analysis

We qualitatively validated the activation map to verify the effectiveness of the proposed method. In Figure 5, the second column is the result of ResNet50, and the third column is the result of applying only the attention module without uniformity loss. The forth column is the result of the proposed method.

Figure 5a shows the occlusion case with obstacles. The second column of each pair focused on the center of the image without considering obstacles. When comparing the third and fourth columns of each pair, the fourth column extracts the comprehensive features of the object and better avoids occlusion. Figure 5b shows the case where the invisible belongings appear as the camera view changes. In this case, belongings are not a common feature. In the second column, the bag was recognized as an important feature in the third row, although the bag was not visible in the front views. In the third and forth columns, we focused on objects other than the bag. Figure 5c is a large pose variation case, where the proposed method extracted suitable features to fit the pose. In addition, in all cases, the fourth column of each pair extracted more accurate and diverse features than the third column.

## 5. Conclusions

In this paper, we presented a novel attentive learning-based Siamese network to extract deterministic features. The proposed network consists of the Siamese network with channel attention module and the encoder-decoder attention module. The channel attention module computes the importance of the feature maps. The encoder-decoder attention module is responsible for seeing whole body rather than local parts. Furthermore, we proposed the uniformity loss utilizing the characteristics of these attention modules. Uniformity loss helps to focus on more deterministic regions and robust on pose variation and occlusion problems. Extensive experiments show that the proposed network compares favorably to the state-of-the-art methods for person re-identification, both in terms of qualitative comparison on various datasets and in terms of quantitative comparison on various metrics.

## Figures and Tables

**Figure 1 sensors-20-03603-f001:**
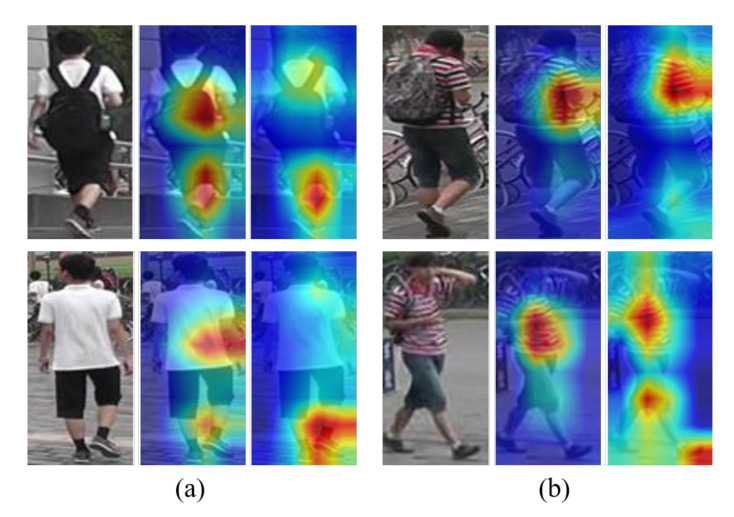
Visualization of the activation maps on the Market1501 dataset. Each set of triplet images respectively represent an input image, activation maps of pre-trained ResNet50 and Ours for two persons (**a**,**b**).

**Figure 2 sensors-20-03603-f002:**
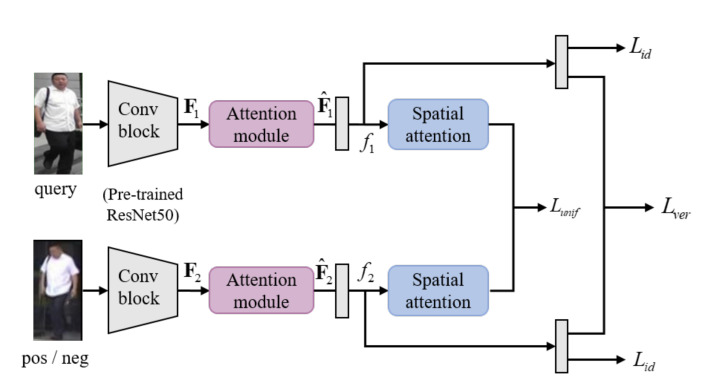
The overall architecture of proposed method.

**Figure 3 sensors-20-03603-f003:**
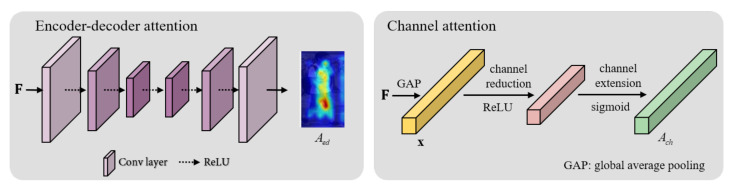
The encoder-decoder attention and channel attention modules.

**Figure 4 sensors-20-03603-f004:**
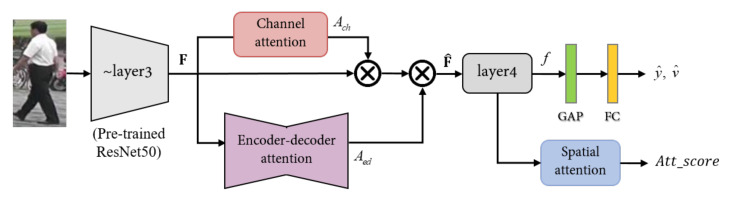
The pipeline of the proposed method. F is the feature map after Resnet50 conv3, $A_{c}$ is the attention map of channel attention module, and $A_{ed}$ is attention map of encode-decode attention module. $z_{i}$ is a feature vector and is used to predict ID and verification. Attention score is compare with positive and negative pairs by Siamese network. GAP and FC denote the global average pooling and fully connected layer.

**Figure 5 sensors-20-03603-f005:**
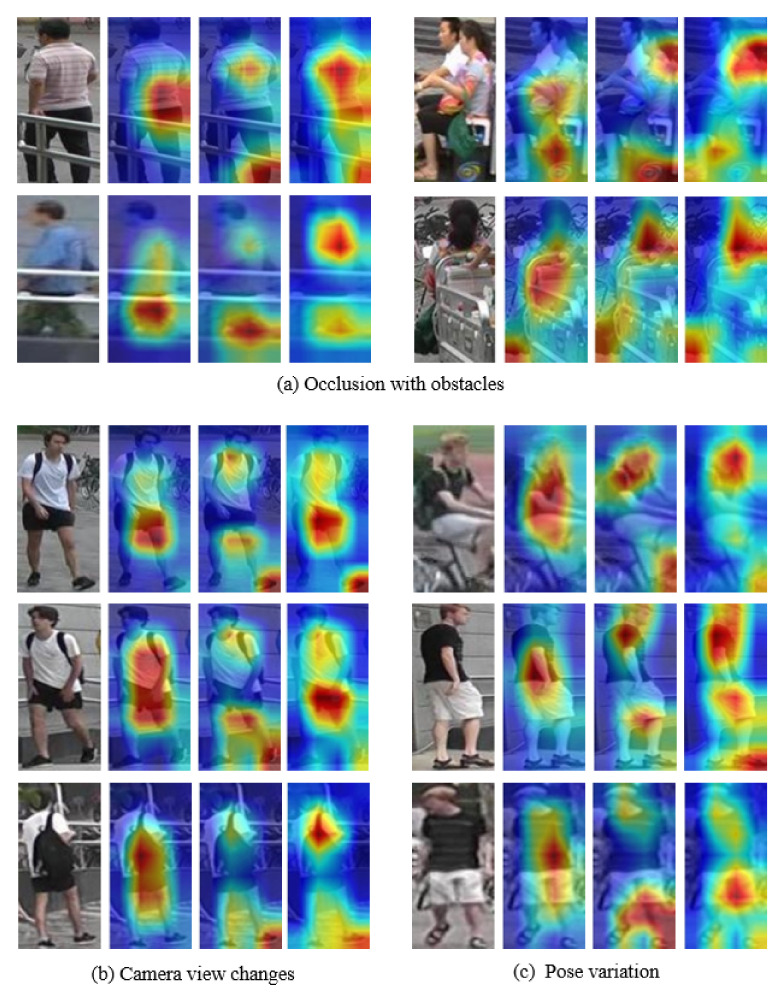
Visualized examples of comparing the proposed method and the others on Market1501 dataset. 1st column is input images, 2nd column is ResNet50, 3rd column is applied only the attention module, and 4th column is the proposed method, respectively.

**Table 1 sensors-20-03603-t001:** Network architecture table of proposed method.

Name	Size	Backbone	Attention Module
Input	128 × 256		
Conv1	64 × 32	7 × 7, 64, stride 2 max pool, 3 × 3, stride 2	
Layer1	64 × 32	$\left[\begin{array}{c} 1\times 1,64\\ 3\times 3,64\\ 1\times 1,256 \end{array}\right]$× 3	
Layer2	32 × 16	$\left[\begin{array}{c} 1\times 1,128\\ 3\times 3,128\\ 1\times 1,512 \end{array}\right]$× 4	
Layer3	16 × 8	$\left[\begin{array}{c} 1\times 1,256\\ 3\times 3,256\\ 1\times 1,1024 \end{array}\right]$× 6	
Channel attention	1 × 1		global average pool 1 × 1, 1024 fc, [1024, 64] fc, [64, 1024]
Encoder	16 × 8		$\left[\begin{array}{c} 3\times 3,256,ReLU\\ 3\times 3,256,ReLU\\ 2\times 2,256,ReLU \end{array}\right]$
Decoder	16 × 8		$\left[\begin{array}{c} 2\times 2,256,ReLU\\ 3\times 3,256,ReLU\\ 3\times 3,1 \end{array}\right]$
Multiple1 Multiple2	16 × 8	ChannelAtt × Layer3 Multiple1 × Decoder	
layer4	8 × 4	$\left[\begin{array}{c} 1\times 1,512\\ 3\times 3,512\\ 1\times 1,2048 \end{array}\right]$ × 3	
Spatial attention	1 × 1		1 × 1, 2048
		global average pool 1 × 1, 2048	

**Table 2 sensors-20-03603-t002:** Comparison with state-of-the-art person ReID methods on the Market1501 dataset.

Market1501				
**Dataset**	**Single Query**		**Multi Query**	
**Metric**	**Rank-1**	**mAP**	**Rank-1**	**mAP**
XQDA [11]	43.8	22.2	54.1	28.4
SCS [46]	51.9	26.3	-	-
DNS [47]	61.0	35.6	71.5	46.0
CRAFT [48]	68.7	42.3	77.0	50.3
CAN [49]	60.3	35.9	72.1	47.9
S-LSTM [50]	-	-	61.6	35.3
G-SCNN [32]	65.8	39.5	76.0	48.4
SVDNet [51]	82.3	62.1	-	-
MSCAN [45]	80.3	57.5	86.8	66.7
HA-CNN [30]	91.2	75.7	93.8	82.8
Ours (ResNet50)	91.3	79.2	94.1	85.3
Ours (VGG16)	89.3	73.3	92.8	81.0

**Table 3 sensors-20-03603-t003:** Comparison with state-of-the-art person ReID methods on the CUHK03 dataset.

CUHK03				
**Dataset**	**Detected**		**Labeled**	
**Metric**	**Rank-1**	**mAP**	**Rank-1**	**mAP**
BoW + XQDA [52]	6.4	6.4	7.9	7.3
LOMO + XQDA [11]	12.8	11.5	14.8	13.6
IDE-R [42]	21.3	19.7	22.2	21.0
IDE-R + XQDA [42]	31.1	28.2	32.0	29.6
PAN [53]	36.3	34.0	36.9	35.0
DPFL [48]	40.7	37.0	43.0	40.5
HA-CNN [30]	41.7	38.6	44.4	41.0
MLFN [54]	52.8	47.8	54.7	49.2
CASN [33]	57.4	50.7	58.9	52.2
Ours (ResNet50)	58.9	52.6	62.6	57.7
Ours (VGG16)	52.7	48.4	46.9	42.2

**Table 4 sensors-20-03603-t004:** Comparison with state-of-the-art person ReID methods on the DukeMTMC-reID dataset.

DukeMTMC-ReID		
**Metric**	**Rank-1**	**mAP**
BoW + KISSME [52]	25.1	12.2
LOMO + XQDA [11]	30.8	17.0
ResNet50 [20]	65.2	45.0
JLML [55]	73.3	56.4
SVDNet [51]	76.7	56.8
HA-CNN [30]	80.5	63.8
Ours (ResNet50)	80.7	65.5
Ours (VGG16)	78.0	61.4

**Table 5 sensors-20-03603-t005:** Efficiency of the proposed method on the Market1501 dataset. The attention module and uniformity loss are denoted AM and UL, respectively.

Dataset	Market1501			
**Metric**	**Rank-1**	**Rank-5**	**Rank-10**	**mAP**
ResNet50-Basel. [34]	88.1	95.0	96.8	71.2
BesNet50 + AM	89.7	96.2	97.4	76.7
Ours(ResNet50 + AM + UL)	91.3	96.9	98.2	79.2
VGG16-Basel. [34]	85.3	94.5	96.3	68.2
VGG16 + AM	87.2	95.5	97.4	69.2
Ours(VGG16 + AM + UL)	89.2	96.1	97.5	73.3

**Table 6 sensors-20-03603-t006:** Comparison on network architectural change on the Market1501 dataset. Each notation means that the attention module (AM) is located after the corresponding layer. Bold indicates the best performance.

Dataset	Market1501			
**Metric**	**Rank-1**	**Rank-5**	**Rank-10**	**mAP**
layer1-AM	89.3	96.3	97.4	74.6
layer2-AM	90.7	96.7	98.0	78.2
layer3-AM	91.3	96.9	98.2	79.2
layer4-AM	88.7	96.1	97.4	75.4

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
