# Peer review of "Uniformity Attentive Learning-Based Siamese Network for Person Re-Identification"

_sensors, 2020, doi:10.3390/s20123603_

Round 1

Reviewer 1 Report

In this article, authors presented a novel attentive learning-based siamese network to extract common and distinctive features. Furthermore, the proposed network emphases with channel attention module which computes the important feature maps and the encoder-decoder attention module is respo

nsible for seeing whole body rather than patches.

The overall idea is convincing, but there are some suggestions that need to be addressed carefully. Therefore, I recommend major revision for this article with the following comments, which can help improving the quality of this work.

Major Comments:

  • In the revised manuscript, the authors should calculate “Cumulative Matching Characteristic (CMC)” curve for each dataset and provide implementation details of the proposed network in experimental section.
  • In experimental section, the detailed description of “Evaluation Metrics” is missing, I suggest discussing all the utilized metrics in detail.
  • The authors should calculate the time complexity of the proposed model and discuss its efficiency for real-life implementation.
  • The proposed network consists of two modules i.e., Channel attention and Encoder-Decoder attention. I suggest authors to deliver more details about Channel attention.

Minor Comments:

  • There are many typos in the current version of this manuscript. I recommend deep proofreading to authors. Some main typos are given below:

Incorrect dataset name on “line no 147”

Space issue on “line no 29”

Should define all abbreviation in first appearance “SQ, MQ,”

Reference is missing on “line no 62”

  • The article structure paragraph is missing which should be included in the last paragraph of the introduction section. Please follow the following paper.

 “Deep Learning Assisted Buildings Energy Consumption Profiling Using Smart Meter Data

Reviewer 2 Report

This paper presents a person re-identification network based on the Siamese network and the attention mechanism. Although the method is feasible, and the results are positive, there are still some points should be explained further.

  1. The background of your study should be enriched.
  2. I think the authors should describe the differences between their model and the method proposed in the following literature.

“Guo, Hao, Kang Zheng, Xiaochuan Fan, Hongkai Yu, and Song Wang. "Visual attention consistency under image transforms for multi-label image classification." In Proceedings of the IEEE Conference on Computer Vision and Pattern Recognition, pp. 729-739. 2019.”

  1. More assessment criteria should be selected to validate your method.
  2. The experimental section is weak. Not only the experimental settings but also the results discussion is explained unclearly.
  3. How about the performance when the backbone network is changed?

Reviewer 3 Report

This manuscript describes a novel approach for re-identification of persons based on Convolutional Neural Networks.

The following points summarize my observations about the manuscript and possible revisions:

  1. In section 2, page 2, line 60, the cited manuscript of “Sermenet et Al.” is not included in the references.
  2. In section 3, the figure 2 is not completely clear. It and its represented quantities should be described in the text. An explanation of the input and output quantities of each block would be useful. The block called “conv” is not defined.
  3. In section 3.1, there are some quantities not properly defined, such as x, y, z, Aed. The dimensions of the involved quantities and layers should be indicated, possibly in a schematic representation, such as figure 3. The schematic representation for the “channel attention module” is missing; it is present only for the “encoder-decorder module” (figure 3).  The mathematical expression of the “encoder-decorder module” is missing; it is present only for the “channel attention module”. A schematic representation with all involved quantities (e.g. x, y, z, Aed, Ac) would be useful.
  4. In section 3.1, the relationship between the feature map F and the feature vector z and the relationship between the feature vector z and the label y are not defined; both the mathematical expression and the schematic representation would be useful.
  5. Section 3.2 deserves a better description. Where do the quantities zq, zp, zn come from? What are the products “*”? How are evaluated the predicted outputs v? A schematic representation and more precise mathematical expressions of this section would be useful.
  6. In sections 3.1 and 3.2, the different main purposes of the two modules “encoder-decorder module” and “channel attention module” should be better explained.
  7. Considerations similar to previous sections 3.1 and 3.2 are valid also for section 3.3. All the involved quantities should be clearly defined and illustrated in a schematic representation. Also the mathematical expressions are useful to understand and apply the proposed method; particularly the expressions for the attention scores s should be added. All the quantities and symbols illustrated in the figures should be introduced and explained (e.g. GAP, FC in figure 4).
  8. In section 3.4, a known reference for ResNet50 should be provided.
  9.  The manuscript completely lacks a section (e.g. 3.5 or better an added new section 4) that deals in detail with the training step of the proposed CNNs. Particularly, the authors should describe how the labels (e.g. y in section 3.1, v in section 3.2) are assigned to the images used in the training set to evaluate the loss functions. Also a description of how the training sets are selected should be added.
  10. Some minor English language improvements are needed.

Recommendation

The proposed approach is promising and interesting. However, the manuscript should be improved following the listed observations. I recommend requiring major revisions.

Round 2

Reviewer 1 Report

Authors addressed all my concerns except one major comment:

"The authors should calculate the time complexity of the proposed model and discuss its efficiency for real-life implementation".

It is very important to consider this comment. Without its possible applicability in the real-world, I think, the impact will be limited.

Reviewer 2 Report

All of the issues have been modified, and the quality of the manuscript is enhanced a lot. I suggest the current version can be accepted.

Author Response

We further revised the submission according to the comments of other reviewers.

Reviewer 3 Report

The modifications introduced by the authors improve the manuscript. I have only one remaining observation that should be considered by the authors: in section 3, the description of Fig.2 is still missing.

I recommend to accept the manuscript requiring minor revisions.
